

# Algorithm for vertical distribution of boundary layer aerosol components in remote sensing data

Futing Wang[1,2], Ting Yang[1,3]*, Zifa Wang[1,2,3], Haibo Wang[1,2], Xi Chen[1,2]

[1]State Key Laboratory of Atmospheric Boundary Layer Physics and Atmospheric Chemistry, Institute of Atmospheric Physics, Chinese Academy of Sciences, Beijing, 100029, China.
[2]College of Earth and Planetary Sciences, University of Chinese Academy of Sciences, Beijing, 100049, China.
[3]Center for Excellence in Regional Atmospheric Environment, Institute of Urban Environment, Chinese Academy of Sciences, Xiamen, 361021, China.

*Correspondence to*: Ting Yang (tingyang@mail.iap.ac.cn)

**Abstract.** The vertical distribution of atmospheric aerosol components is vital to the estimation of radiation forcing and the catalysis of atmospheric photochemical processes. Based on the synergy of ground-based lidar and sun-photometer, this paper developed a new algorithm to get the vertical mass concentration profiles of fine aerosol components for the first time. The sky radiance at multiple scatter angles, the total optical depth (TOD) at 440, 675, 870, and 1020 nm, and the lidar signals at 532 nm and 1064 nm were applied to retrieve the aerosol properties. Besides, the internal mixing model and normalized volume size distribution model were constructed, according to the absorption and water-solubility of aerosol components, to separate the profiles of black carbon (BC), water-insoluble organic matter (WIOM), water-soluble organic matter (WSOM), ammonium nitrate-like (AN), and fine aerosol water content (AW). The results showed a reasonable vertical distribution of aerosol components compared with in situ observations and reanalysis data. The estimated and observed BC concentration matched well with a correlation coefficient up to 0.91, while there was an evident overestimation of OM (NMB=0.98). And the retrieved AN concentrations were closer to the simulated results (the correlation coefficient of 0.85), especially in the polluted condition. The correlations of BC and OM were weaker relatively, with a correlation coefficient of about 0.5. Besides, the uncertainties caused by input parameters (i.e. RH, volume concentration, and extinction coefficients) were assessed by Monte Carlo method. AN and AW had smaller uncertainties at higher RH. In this paper, the algorithm was also applied to the remote sensing measurements of Beijing and two typical cases were presented. Under the clean condition with low RH, there were comparable AN and WIOM but peaking at different altitudes. While in the polluted case, AN was dominant and the maximum mass concentration occurred near the surface. We expected the algorithm can provide a new idea for lidar inversion and promote the development of aerosol components profiles.

## 1 Introduction

Atmospheric aerosols play a key role in the radiation budget and energy balance (Andrews and Forster, 2020; Hasekamp et al., 2019). The aerosols with different optical and physical properties have diverse radiative forcing effects (Boucher et al.,



2013). For example, the soot dominated by black carbon has the most significant effect on cloud cover and precipitation due to its strong absorption (Gu et al., 2006; Xia, 2014), while the negative radiative forcing of sulfate and nitrate is more prominent (Myhre et al., 2013). Especially, different aerosol components can even cause opposite radiation changes vertically (Jiang et al., 2013). On the other hand, aerosols can affect the atmospheric oxidation capacity by changing the

photolysis rate of trace gases (Bian et al., 2003; Lou et al., 2014; Xing et al., 2017). Liao et al. (1999) once found that nonabsorbent aerosols generally enhanced photolysis rates, contrary to the soot aerosols. What's more, the ability of aerosols to reduce the photolysis frequency of $O_3$ decreases with altitude on a regional scale (Li et al., 2011). Therefore, the vertical distribution of atmospheric aerosol components is of vital importance to reduce the uncertainty of radiation forcing estimation and understand the impact of haze on atmospheric photochemical processes.

At present, there are many studies focusing on the aerosol components on the ground (Han et al., 2015; Huang et al., 2014; Zhao et al., 2013). However, the ways to get the aerosol components in the atmosphere are finite. Although the field campaigns often launched aircrafts (Chen et al., 2009; Zhang et al., 2009) or tethered balloons (Li et al., 2015; Ran et al., 2016) to detect the atmospheric structure, there are still many limitations in the resolution and representation due to the restricted aircraft control. In such a situation, continuous remote sensing technology with high temporal resolution, such as

sun-photometer and lidar, provides a powerful tool for the identification of aerosol components. Besides, the establishment of uniquely distributed ground-based networks, e.g., AERONET (Dubovik et al., 2002; Dubovik et al., 2000), AD-Net (Shimizu et al., 2017; Nishizawa et al., 2017), and MPLNET (Chew et al., 2013; Huang et al., 2011), also improves the spatial detection resolution.

So far, the algorithms using instantaneous remote sensing measurements to retrieve atmospheric aerosol components have

been greatly developed. The available aerosol parameters from sun-photometer make it possible to distinguish the components. Schuster et al. (2005) proposed a three-component model constrained by the refractive index to infer black carbon, ammonium sulfate, and water. Subsequently, the absorbing organic carbon and dust were supplemented to the model by Arola et al. (2011) and Wang et al. (2012) respectively. By joint use of the refractive index, single scattering albedo, sphericity, and other measurements, Van Beelen et al. (2014) and Xie et al. (2017)  greatly increased the identifiable aerosol

components. Besides, internal mixing and hygroscopic growth of aerosols were also considered in the algorithms to reproduce the real state (Schuster et al., 2016; Zhang et al., 2018a; Zhang et al., 2020).

However, the aerosol vertical distribution is not available from the sun-photometer, which can be made up by the ground-based lidar with its vertical resolution of meters. Burton et al. (2012) and Groß et al. (2011) found the characteristics of different aerosol types in lidar parameters, proving the feasibility of lidar. Nishizawa et al. (2007) took use of dual-

wavelength elastic lidar but only separated water-soluble aerosol from dust or sea salt. After that, the application of Raman lidar and more wavelengths made it possible to get the profiles of sea salt, soot, dust, and water-soluble aerosols (Nishizawa et al., 2011; Nishizawa et al., 2017; Hara et al., 2018). Mamouri and Ansmann (2017) refined the fine and coarse dust and separate them from maritime and anthropogenic aerosols based on a polarization/Raman lidar. However, limited by the available lidar information, there has been no breakthrough in the aerosol types identified by lidar measurements. Therefore,



how to use limited lidar channels to distinguish more atmospheric aerosol components is what we need to investigate. And considering the advantages and limitations of sun-photometer and lidar, it may be a good choice to combine them.

In this study, a new algorithm to get the vertical profiles of fine aerosol components based on the combination of ground-based Mie lidar and sun-photometer is proposed for the first time, which includes black carbon (BC), water-insoluble organic matter (WIOM), water-soluble organic matter (WSOM), ammonium nitrate-like (AN), and fine aerosol water

content (AW). The details about the algorithm and the measurement data applied to the algorithm will be described in Sect. 2. And Sect. 3 will present the evaluation, uncertainty analysis, and application of our algorithm to confirm the validity of inversion results.

## 2 Methodology and data

### 2.1 Methodology

#### 2.1.1 Aerosol microphysical characteristics

Different size distribution and complex refractive indexes make it possible to separate aerosol components by remote sensing. Exactly as Table 1, BC has the largest CRI than others at different wavelengths, which indicates its strong optical absorption (Mueller et al., 2007; Burton et al., 2012). On the contrary, AN (denoting the inorganic salt such as nitrate and sulfate) is mainly characterized by scattering with the smallest CRI (Zhang et al., 2012; Xu and Penner, 2012), except for AW.

Generally, AW content directly depends on the hygroscopic AN at a certain ambient RH, especially in heavy haze episodes (Zhang et al., 2015). Here, OM was considered as two components: WIOM and WSOM, and has a little spectral change, which is significantly different from BC in the water-insoluble matter. While in water-soluble ones, hygroscopicity is the key to distinguishing WSOM and AN. According to the summary in Zhang et al. (2018a), the growth factors of inorganic salts are all above 1.5, much larger than that of WSOM. Thus, it is considered that the aerosol hygroscopicity only comes from

AN rather than WSOM in this algorithm to separate them.

**Table 1.** Microphysical parameters of atmospheric aerosol components used in this paper, including complex refractive index (CRI), mean radius $r_m$ and geometric standard deviation σ of the lognormal distribution, and density. The relevant literature is given in the last row of the table.

|  |  | BC | WIOM | WSOM | AN | AW |
|---|---|---|---|---|---|---|
| CRI | 532 nm | 1.95-0.79i | 1.56-0.06i | 1.53-0.003i | 1.41-2.3e-3i | 1.33 |
|  | 1064 nm | 1.95-0.79i | 1.54-0.001i | 1.53-0.001i | 1.40-6.8e-3i | 1.33 |
| $r_m$ (μm) |  | 0.095 | 0.126 | 0.126 | 0.17 | - |
| σ |  | 1.8 | 1.49 | 1.49 | 2 | - |


| $\rho$ (g cm$^{-3}$) | 2.0 | 1.2 | 1.2 | 1.76 | 1.0 |
|---|---|---|---|---|---|
| References | Van Beelen et al. (2014); Ganguly et al. (2009); Schuster et al. (2005) | Kirchstetter et al. (2004); Schuster et al. (2016); Ganguly et al. (2009); Dey et al. (2006) | Arola et al. (2011); Ganguly et al. (2009) | Hess et al. (1998); Ganguly et al. (2009); Van Beelen et al. (2014) | Dey et al. (2006); Schuster et al. (2005); Van Beelen et al. (2014) |


### 2.1.2 Fine mode aerosol properties from GARRLiC

In fact, there have been developed algorithms combining the sun/sky photometer with lidar. For pursuing an even deeper synergy of lidar and sun-photometer, Generalized Aerosol Retrieval from Radiometer and Lidar Combined data (GARRLiC) was created by modification of AERONET algorithms to adapt them for inclusion of lidar data (Lopatin et al., 2013). As a

part of the extensive Generalized Retrieval of Atmosphere and Surface Properties (GRASP), it can get the properties profiles separately for fine and coarse mode particles and has been applied for the characterization of atmospheric aerosols (Lopatin et al., 2013; Tsekeri et al., 2017; Bovchaliuk et al., 2016) and evaluated by air-borne in situ measurements (Benavent-Oltra et al., 2021; Benavent-Oltra et al., 2017). Although GARRLiC was able to quantitatively retrieve aerosol components (Li et al., 2019a), it still stayed on the columnar level and can not get the flexible volume proportion of different components

vertically.

Note particularly that the volume concentration profiles provided by GARRLiC build a bridge between the retrieval of sun-photometer and lidar. In previous lidar algorithms, lidar parameters such as lidar ratio (the ratio of extinction to backscattered coefficients), were employed to avoid the use of volume concentration. But now, due to the accessible volume concentration, complex refractive index and volume size distribution can be directly used to construct the aerosol model in

lidar algorithms based on Mie theory (Bohren and Huffman, 1998). Consequently, the combination of ground-based remote sensing technology not only enriches the inversion output but also provides a new idea for lidar inversion.

However, Mie theory is only applicable to spherical particles, which is in contradiction with the irregular shape of dust aerosols (Mamouri and Ansmann, 2014; Sugimoto et al., 2002). Generally, it's assumed in retrieval algorithms that the aerosol size distribution is bimodal and the dust aerosol is distributed in the coarse mode (Nishizawa et al., 2007; Nishizawa

et al., 2011; Schuster et al., 2016; Xie et al., 2017; Zhang et al., 2018a). Therefore, in this study, we only focus on the fine aerosol components profiles based on the outputs of GARRLiC, which includes the aerosol extinction and volume concentration profiles in the fine mode. Similar to AERONET (Dubovik and King, 2000), the radius of 0.576 μm was used as a separation point in GARRLiC. According to the field experiments, the retrieved fine mode aerosol components, including BC, WIOM, WSOM, and AN, were almost distributed in PM$_1$ (particles with the aerodynamic diameter less than 1

μm) (Liu et al., 2020; Reddington et al., 2013; Zhang et al., 2018b). To some extent, the fine modal truncation radius of 0.576 μm is reasonable for inversion.



### 2.1.3 Aerosol modeling

In the actual atmosphere, the internal mixing of aerosols is very common due to aerosol collision, condensation, and chemical reactions. Generally, Maxwell-Garnett (MG) mixing rule is more appropriate for the mixture of water-insoluble matter embedded in the host environment (Choi and Ghim, 2016; Dey et al., 2006; Schuster et al., 2005). The effective permittivity of the mixture $\varepsilon_{mix}$ can be expressed as follows:

$$\varepsilon_{mix}(\lambda) = \varepsilon_{host} + 3\varepsilon_{host}\left[\frac{\sum_j \frac{\varepsilon_j(\lambda)-\varepsilon_{host}(\lambda)}{\varepsilon_j(\lambda)+2\varepsilon_{host}(\lambda)}f_j}{1-\sum_j \frac{\varepsilon_j(\lambda)-\varepsilon_{host}(\lambda)}{\varepsilon_j(\lambda)+2\varepsilon_{host}(\lambda)}f_j}\right] \qquad j = BC\ and\ WIOM \tag{1}$$

where $f_j$ is the volume fraction of component $j$. Here, host environment represents the mixture of water-soluble matter, including AN, WSOM and AW. $\varepsilon_{host}$ and $\varepsilon_j$ are the effective permittivities of the host environment and insoluble matter respectively, which can be calculated from the corresponding CRIs by Eq. (2).

$$m = \sqrt{\frac{|\varepsilon(\lambda)|+Re(\varepsilon(\lambda))}{2}} + i\sqrt{\frac{|\varepsilon(\lambda)|-Re(\varepsilon(\lambda))}{2}} \tag{2}$$

where $m$ and $\varepsilon$ are the CRI and effective permittivity respectively. For the CRI of the host environment $m_{host}$, which refers to the water-soluble matter, can be obtained by the volume averaged (VA) mixing rule to strengthen the physical constraints between multi-component liquid systems (Zhang et al., 2018a).

$$m_{host}(\lambda) = \frac{\sum_j m_j(\lambda)f_j}{\sum_j f_j} \qquad j = AN, AW\ and\ WSOM \tag{3}$$

where $m_j$ and $f_j$ are the CRI and volume fraction of soluble components respectively. Then the CRI of the aerosol mixture $m_{mix}$ can be acquired by combining Eq. (1)–(3).

In addition to CRI, volume size distribution (VSD) is the other requirement for Mie theory. Here, the normalized VSD of each component can be simulated according to the lognormal distribution parameters in Table 1, which are all in the dry state. Considering the hygroscopicity of AN, the growth factor is introduced to fit the AN normalized VSD under ambient RH (AW is taken into account at the same time). Then, we can model the normalized VSD of aerosol mixture based on the assumed component volume fraction $f_j$ as follows:

$$\frac{dV_N(lnr)}{dlnr} = \sum_{j=1}^{4} f_j \frac{dV_j(lnr)}{dlnr} \qquad j = AN, BC, WIOM, and\ WSOM \tag{4}$$

where $\frac{dV_N(lnr)}{dlnr}$ is the normalized VSD of the aerosol mixture. $\frac{dV_j(lnr)}{dlnr}$ is the normalized VSD of component $j$ and can be expressed by Eq. (5).

$$\frac{dV_j(lnr)}{dlnr} = \frac{1}{\sqrt{2\pi}|ln\sigma_j|}\exp\left[-\frac{1}{2}\left(\frac{lnr-lnr_j}{ln\sigma_j}\right)^2\right] \tag{5}$$

where $\sigma_j$ and $r_j$ are the geometric standard deviation and mean radius of component $j$ respectively, which are listed in Table 1.



Combining the fine mode volume concentration profiles $V(h)$ from GARRLiC, the extinction coefficients at different

wavelengths and levels $\sigma_m(\lambda, h)$ can be modeled according to Mie theory:

$$\sigma_m(\lambda, h) = \int \frac{3}{4r^2} Q_{ext}(\lambda, r, m) \frac{dV(lnr)}{dlnr} dlnr \qquad (6)$$

where $Q_{ext}$ is the Mie efficiency factor, which is related to lidar wavelength, particle size, and CRI (Bohren and Huffman,

1998). $\frac{dV(lnr)}{dlnr}$ can be obtained by $V(h)\frac{dV_N(lnr)}{dlnr}$.

Finally, the residual between modeled extinction $\sigma_m$ and fine mode extinction from GARRLiC $\sigma_c$ is quantified by the

iterative kernel function $\chi^2$ to find the optimal combination of component volume fractions.

$$\chi^2 = \sum_\lambda \frac{(\sigma_m(\lambda,h) - \sigma_c(\lambda,h))^2}{\epsilon_g(\lambda,h)(\sigma_c(\lambda,h))^2} \qquad \lambda = 532, 1064 \; nm \qquad (7)$$

where $\epsilon_g(\lambda, h)$ is the relative fitting residual between lidar measurement and modeled lidar signal from GARRLiC at

different wavelengths, which is added to avoid the interference of the uncertainty resulting from GARRLiC modeling.

Further, the component volume fractions can be transformed to the mass concentrations $M_j(h)$ by the density ($\rho_j$) of aerosol

component $j$.

$$M_j(h) = f_j(h) \times V(h) \times \rho_j \quad j = AN, AW, BC, WIOM, and \; WSOM \qquad (8)$$

### 2.1.4 Microphysical parameterization scheme

The matched number of input parameters and the output aerosol types is the prerequisite for a reasonable aerosol model. Due

to the limitation of lidar wavelengths, the input parameters of lidar are not as many as sun-photometer. Therefore, the aerosol

parameterization scheme should be constructed to establish the relationship between aerosol components, thereby reducing

the number of unknowns. In our algorithm, we separated water-soluble and water-insoluble matter firstly by the

parameterization scheme of Zhang et al. (2018a), which was re-parameterized with relative humidity (RH) based on Schuster

et al. (2009). The volume ratio of water-insoluble to water-soluble matter can be expressed as follows:

$$\frac{f_i}{f_s} = \varphi(RH) \int \varepsilon(D) dD \qquad (9)$$

$$\varphi(RH) = 5.74(1 - RH)^3 + 0.01 \qquad (10)$$

$$\varepsilon(D) = \varepsilon_0 + \varepsilon_v * \exp\left[-(\frac{\log(D/d_0)}{\sigma_{log}})^2\right] \qquad (11)$$

where $f_i$ and $f_s$ are the water-insoluble and water-soluble volume fractions respectively. $\varphi(RH)$ is the re-parameterized part

of the function with RH. $\varepsilon(D)$ is the climatological function of water-soluble volume fraction and $D$ is the aerosol diameter.

$\varepsilon_0$, $\varepsilon_v$, $d_0$ and $\sigma_{log}$ are the average fitting parameters in Kandler and Schuetz (2007), which can represent the general aerosol

properties.

For the water-soluble matter, we assumed that AN was the only hygroscopic component as mentioned in Sect. 2.1.1. For

enhancing the interaction between AN and AW, the relationship between solute mass concentration and water activity was



applied in our algorithm, which was investigated in Tang (1996). And the volume ratio of AN to AW can be obtained by combining the Equation (12)–(15):


$$a_w = 1 + \sum_{k=1}^{4} C_k x^k \tag{12}$$

$$RH = a_w/100 \tag{13}$$

$$\rho_s = 0.9971 + \sum_{k=1}^{4} A_k x^k \tag{14}$$

$$\frac{f_{AN}}{f_{AN}+f_{AW}} = x \frac{\rho(x)}{\rho(100)} \tag{15}$$

where $a_w$ is the water activity, which can be approximately regarded as RH due to the lower curvature effect (Tang, 1996).

$\rho_s$ is the density of solution and $x$ is the weight percent of AN. $C_k$ and $A_k$ are the polynomial coefficient of ammonium nitrate from Tang (1996), which is considered as the representive of inorganic salt. $f_{AN}$ and $f_{AW}$ are the volume fractions of AN and AW respectively. With that, the growth factor (GF) of AN can also be acquired, which palys a vital role in the aerosol normalized volume distribution model of Sect. 2.1.3.

$$GF(RH) = \frac{r_{wet}(RH)}{r_{dry}} = \sqrt[3]{\frac{f_{AN}+f_{AW}}{f_{AN}}} \tag{16}$$

where $r_{dry}$ is the dry particle radius; $r_{wet}$ is the particle radius under the ambient RH.

Based on the above relationship, only two unknowns are enough to get what we wanted. In our algorithm, we iteratively changed the volume fractions of WIOM and WSOM. Correspondingly, that of AN, AW, and BC can be obtained. What's more, we also constrained the relationship between WSOM and WIOM to ensure the reliability of inversion, which has been applied in Zhang et al. (2018a) according to the statistics of observation experiments.

In summary, Fig. 1 gives the flowchart of our algorithm proposed in this study. If the volume fractions of BC, WIOM, WSOM, AN, and AW are given, the extinction coefficient can be calculated by the constructed aerosol model. Through multiple iterations, the optimal combination of volume fractions will be found.





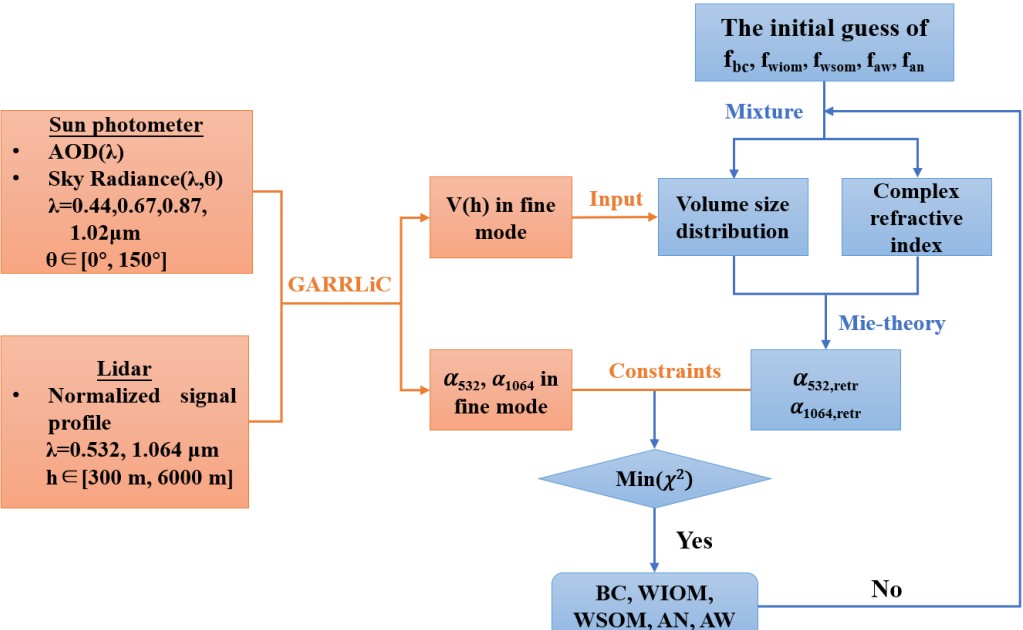

**Figure 1.** Flowchart of the algorithm proposed in this paper.

## 2.2 Measurement data

### 2.2.1 The input data of GARRLiC

The input data of GARRLiC consist of sun-photometer sun and sky radiance, and lidar signals. Here, the sun-photometer measurements were from the Beijing station (39.977° N, 116.381° E) of the AERONET (Aerosol Robotic Network, https://aeronet.gsfc.nasa.gov/) in February of 2021. The sky radiance (raw almucantar with 26 scattering angles) and the version 3 level 1.5 product (i.e. automatically cloud cleared but may not have final calibration applied) of total optical depth (TOD) at 440, 675, 870, and 1020 nm were applied to drive GARRLiC. Besides, the AERONET products of fine AOD and fine volume concentration were employed to validate the outputs from GARRLiC.

For the available sky radiance sequence, the correlative lidar signals data were chosen from a dual-wavelength elastic lidar in the corresponding ±15 min time window, which was set up on the roof of a 28 m high building in the tower of the Institute of Atmospheric Physics at the Chinese Academy of Sciences (39.976° N, 116.378° E). The normalized lidar signals at 532 nm and 1064 nm were used to run the GARRLiC with the sun-photometer data. In advance, the lidar signals were averaged for 15 min and computed for 60 log-spaced heights between 150 m and 6000 m above the ground to avoid the instrumental error just as the Equation (17).

$$P_k' = P_k / \int_{Z_{min}}^{Z_{max}} P_k \, dZ \qquad (17)$$



Where $P_k{}'$ is the normalized lidar signal and $P_k$ is the raw averaged lidar signal. $Z_{max}$ and $Z_{min}$ represents the upper and lower height limit, respectively. Besides, for the accuracy of GARRLiC, the cases with the relative residual larger than 15 % in the inversion process have been eliminated according to Benavent-Oltra et al. (2021). Consequently, there were 133 retrievals remaining in February of 2021.

**2.2.2 Relative humidity data**

The vertical profile of relative humidity (RH) data used in our algorithm was interpolated linearly from the European Centre for Medium-Range Weather Forecast (ECMWF) Reanalysis v5 (ERA5) hourly data from 1000 hPa to 300 hPa, which has been verified by the sounding data from the University of Wyoming (http://weather.uwyo.edu/upperair/bufrraob.shtml) in Fig. S1 of Supplement.

**2.2.3 Components data**

The mass concentrations of aerosol components near the surface on 8–15 February 2021, including water-soluble inorganic salt, BC, and organic carbon (OC), were provided by the China National Environmental Monitoring Center to validate the retrieved components results. Besides, the Nested Air Quality Prediction Model System (NAQPMS), a three-dimensional chemistry transport model developed by the Institute of Atmospheric Physics (IAP) (Li et al., 2012), was also employed to verify the reliability of estimated component profiles. The meteorology field was provided by the Weather and Forecasting

model (WRF), which is driven by Final Analysis data (FNL) from the National Centers for Environmental Prediction (NCEP). And the outputs of NAQPMS used in this paper have been assimilated through the Parallel Data Assimilation Framework (PDAF) system, which has a fairly good correlation with measurements (Wang et al., 2022).

**3 Results and discussion**

**3.1 Validation**

**3.1.1 Evaluation of the outputs from GARRLiC**

Since GARRLiC provides the input and constraints for our algorithm, whether the GARRLiC outputs are reliable directly determines the accuracy of the component inversion. Therefore, the GARRLiC outputs have to be validated based on the products of AERONET, which is widely used in the validation of remote sensing results (Che et al., 2009). Due to the unavailable volume concentration profile of AERONET, Fig. 2a presents the comparison of fine columnar volume

concentration between GARRLiC and AERONET. It's clear that the correlation coefficient (R) can be up to 0.94 and the Root Mean Square Error (RMSE) was only 0.017. The Mean Percent Error (MPE), which is the average percent of error from the truth, was about 42 %. This deviation was acceptable since the estimated uncertainty for CRI in the Level 2 AERONET products is about 50 % (Dubovik et al., 2000). Moreover, the extinction coefficients from GARRLiC were also



compared with the results retrieved by the Fernald method (Fernald, 1984) with the lidar ratio of 50 sr (Wang et al., 2020).

From Fig. 2b we can see that the two results were highly consistent and R was close to 1. Figure 2c shows the vertical distribution of extinction coefficients from GARRLiC and lidar. Obviously, the extinction average and standard deviation profiles of the two almost coincided, confirming the validity of the GARRLiC outputs. In fact, the extinction profiles from GARRLiC depend directly on the fine mode AODs and the aerosol vertical profiles (unit: km$^{-1}$), which are retrieved by lidar signal. Therefore, we validated the fine mode AOD and fitting lidar signal with AERONET and lidar signal measurements,

respectively. As shown in Fig. 3, not only the fine mode AOD but also the fitting lidar signal was in good agreement with their respective reference, with the R greater than 0.99. And the total MPEs of fine extinction at two wavelengths were both about 14 %, largely dependent on the fine mode AOD due to the little error in vertical lidar signal fitting. All the above analysis indicates that the fine volume concentration and extinction profiles from GARRLiC are reliable enough to drive the component retrieval.

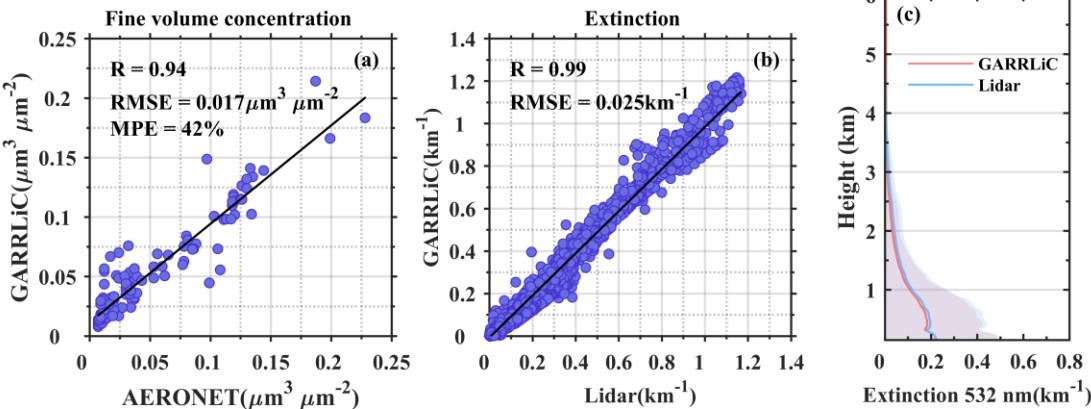

**Figure 2.** (a) The comparison of fine volume concentration between GARRLiC and AERONET; (b) The comparison of extinction coefficient at 532 nm between GARRLiC and lidar; (c) The averaged vertical extinction profiles from GARRLiC and lidar in February of 2021. The shadows with different colors represent the standard deviation of extinction profiles from GARRLiC and lidar.

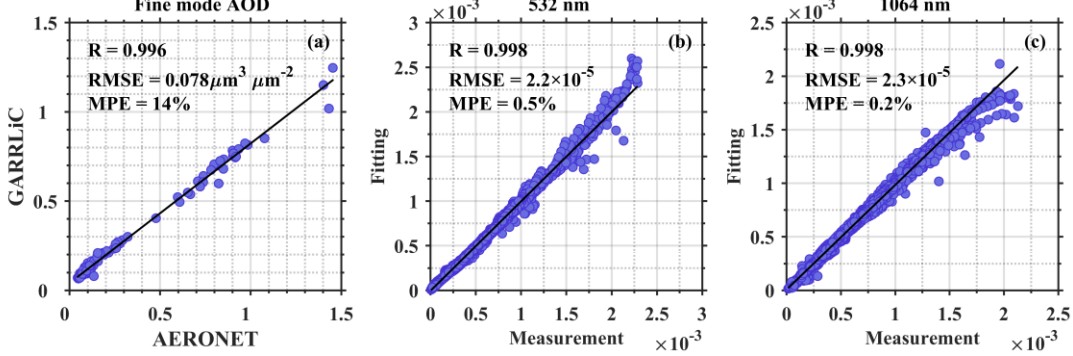





**Figure 3.** (a) The comparison of fine mode AOD between GARRLiC and AERONET; The comparison of lidar signal between the fit from GARRLiC and measurement at (b)532 nm and (c) 1064 nm.

### 3.1.2 Comparison with surface observations

In order to validate the estimated mass concentration of components, the comparison with observations on 8–15 February 2021 is presented here. During the test experiment period, the number of samples was limited by the availability of AERONET data. As shown in Fig. 4a, the fine volume concentration between GARRLiC and AERONET matched well, with a correlation coefficient of 0.89. The Normalized Mean Bias (NMB) of 0.031, an index to measure the deviation from true values (Wang et al., 2021), indicated the credible results from GARRLiC. But there was still a slight underestimation at high aerosol loading. On 11–12 February 2021, the RH dropped from the peak value companied with decreased extinction coefficients (Fig. 4b–c). And the RH in the experiment period changed from 20 % to 70 %, which was enough to reflect the general atmospheric situation.

Due to the lack of observed component profile, the observed mass concentrations of water-soluble inorganic salt, BC, and organic carbon (OC) near the surface were used to verify the remotely sensed results preliminarily. The estimated components from remote sensing at 150 m were employed for the verification. For comparable components, we selected the mass concentration of OC multiplied by the conversion factor of 1.7 (Burki et al., 2020) as the observed organics to compare with OM (WIOM+WSOM) due to the limited observation. And the sum of the water-soluble inorganic salt was used to compare with AN. Figure 5 gives the comparable results of AN, BC, and OM between observation and retrieval results at the available time. An encouraging coherence in the variation trend of AN between estimation and observation (R=0.67) was found although there was an underestimation on 12 February. Besides, there was a better consistency between the estimated and observed BC. The correlation coefficient can be up to 0.91. However, the overestimation of BC was obvious on 12 February, which was just the opposite of AN. This deviation can be attributed to the decreasing RH from 11 February to 12 February, which influences the parameterization schemes as mentioned in Sect. 2.1.4. Moreover, when the extinction coefficients changed little (Fig. 4b), the small fine volume concentrations after 12 February also had responsibility for the error, which led to the underestimated total mass concentration relative to observation. As shown in Fig. 5c, the overestimation of OM was evident, and the mass concentrations of WSOM were closer to the observation. We should note that the components in the remote sensing models are not equivalent to the concepts in chemical research (Li et al., 2019b), which is the primary error of comparisons. On the other hand, there must be differences in the mass concentration of aerosol between the surface and 150 m due to the influence of the atmospheric mixing state and emission sources of different aerosol components.





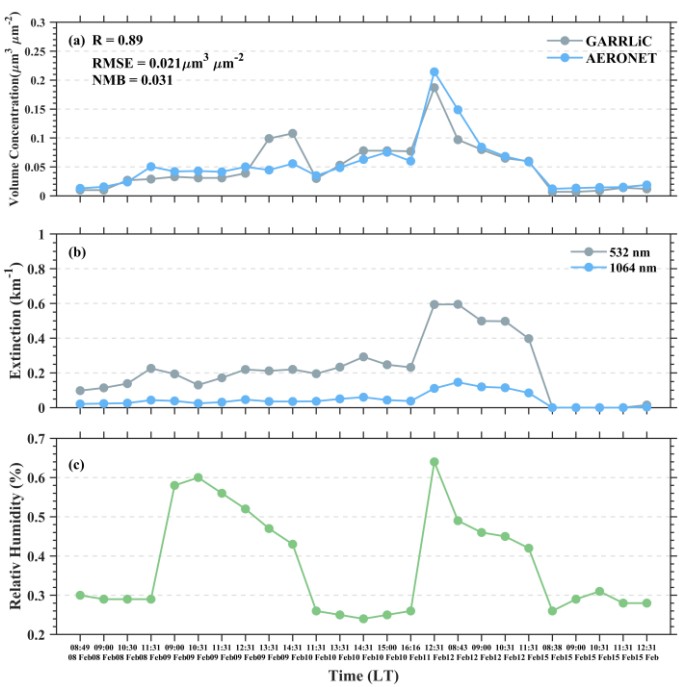

**Figure 4.** (a) The fine volume concentration from GARRLiC and AERONET; (b) The fine mode extinction at 532 nm and 1064 nm; (c) The RH from ERA5 at the available time on 8–15 February 2021.

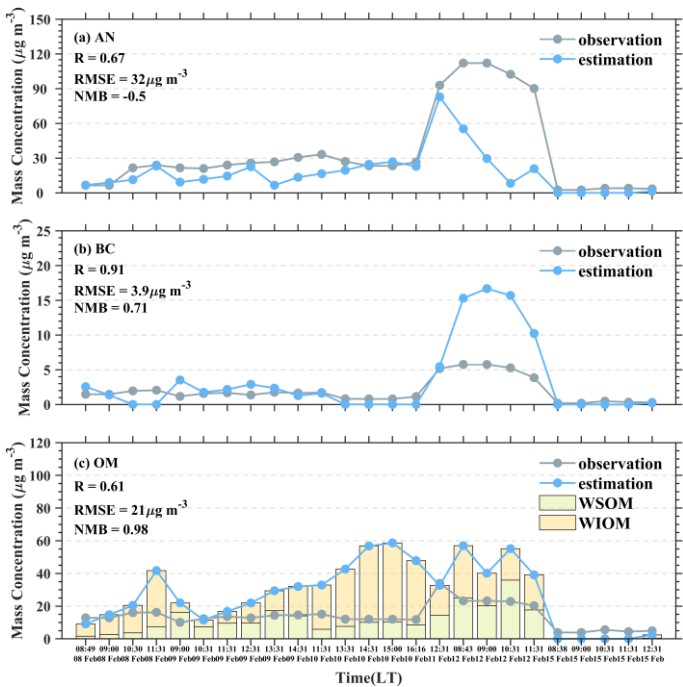

**Figure 5.** The comparisons between observation and estimation results from remote sensing of (a) AN; (b) BC; (c) OM at available time on 8–15 February 2021. The correlation coefficient (R), Root Mean Square Error (RMSE), and Normalized Mean Bias (NMB), which is



defined as $\sum(con_{obs} - con_{est})/\sum con_{obs}$, are presented. The subscript of "obs" and "est" means the observation and retrieval, respectively.

### 3.1.3 Verification of estimated vertical profiles

Similar to the comparisons with surface observations, the mass concentration of OC multiplied by 1.7 was chosen to compare with OM, and the sum of the nitrate, sulfate, and ammonium salt was used to compare with AN. The estimated AN

from remote sensing had the best correlation with that from NAQPMS (R=0.85 in Table 2), and there was a slight underestimation (NMB=-0.19). While the correlations of BC and OM were weaker relatively, with a correlation coefficient of about 0.5, which corresponded to the relationship of OM in Zhang et al. (2018a). The deviation can be explained by the different input RH data of our algorithm and NAQPMS. What's more, the differences in the results from two different principles are reasonable. After all, our classification of aerosol components is based on their optical characteristics. Here, in

order to evaluate the performance of vertical profiles, we present two cases with lower and higher aerosol loading in Fig. 6. It can be seen that the mass concentration profiles of aerosol components from remote sensing and NAQPMS were comparable. Under the relatively clean condition, the mass concentrations of OM were higher than that of AN, and the estimated OM had a similar vertical distribution to that from NAQPMS with the smallest deviation. There were fluctuations in both estimated and simulated AN profiles below 1 km. Subsequently, the local maximum concentration of estimated AN

occurred at 2.5 km, while at about 2 km for simulated AN profiles. The distribution of the two BC profiles was similar. Under the situation with a higher aerosol load (Fig. 6d–f), the estimation of AN performed best according to the simulated AN profile from NAQPMS, but still with a little underestimation. While the overestimation of OM and BC at about 1 km was obvious. It is noteworthy that the vertical distribution of different aerosol components was synchronous in both remote sensing and NAQPMS. Therefore, the vertical patterns of components depend largely on that of total extinction profiles,

which is why the two results from remote sensing and NAQPMS cannot match exactly.

**Table 2.** The correlation coefficient (R), the Root Mean Square Error (RMSE), and the Normalized Mean Bias (NMB) of AN, BC, and OM between remote sensing and NAQPMS are presented.

| | R | RMSE ($\mu g\ m^{-3}$) | NMB |
|---|---|---|---|
| AN | 0.85 | 14.4 | -0.19 |
| BC | 0.54 | 5.2 | -0.18 |
| OM | 0.50 | 15.9 | 0.78 |





**Figure 6.** The averaged mass concentration profiles of (a) AN; (b) OM; (c) BC from 11:00 to 13:00 (LT) on 9 February, 2021; (d)–(f) Same as (a)–(c) but averaged from 9:00 to 12:00 (LT) on 11 February, 2021. The solid lines and dashed lines represent the results from remote sensing and NAQPMS, respectively.

## 3.2 Uncertainty assessment of components estimation

In fact, the uncertainties of component retrieval mainly come from the errors of input parameters, i.e. RH, volume concentration, and extinction coefficients. Among them, RH influences the components estimation indirectly by the parameterization schemes, which are closely related to RH. Zhang et al. (2018a) have discussed the uncertainty of $\frac{f_i}{f_s}$ caused



by $\varphi(RH)$ and the mean error is about 31.6 % when the RH is no more than 85 %. Since the influence of RH on parameterization scheme always exists, here, we take 55 % as the critical point of higher and lower RH to evaluate the uncertainty from input microphysical parameters. In this paper, Monte Carlo method was employed based on the random

generation of input parameters by a Gaussian distribution with the original values and errors as mean and standard deviation, respectively. The error of RH was considered as about 10 % according to the uncertainty from ERA5 (Gamage et al., 2020). And MPEs of the fine volume concentration and extinction coefficient mentioned in Sect. 3.1.1, were applied in the Monte Carlo method. Each input parameter was sampled with 30 iterations at different heights (Mattis et al., 2016). Relative uncertainty was characterized by the ratio of standard deviation to mean values of 30 iteration results.

The uncertainties of AN, AW, WSOM, BC, and WIOM from RH are given in Fig. 7a, with the mean values of 34.5 %, 48 %, 16.5 %, 40 %, and 7 % under the low-RH condition, 24.7 %, 40 %, 57 %, 68.8 %, and 65.8 % under the high-RH condition, respectively. For other parameters, there were similar quantitative relationships of the components estimation uncertainties. The uncertainties of AN and AW at higher RH were smaller than those at lower RH for all parameters. That's because the parameterization scheme described in Sect. 2.1.4 is closer to the actual condition at higher RH (Tang, 1996). On

the contrary, the higher RH made the larger uncertainties for WIOM, BC, and WSOM, which may be due to the increasing error of $\frac{f_i}{f_s}$ caused by $\varphi(RH)$ below the RH of 85 % (Zhang et al., 2018a). As shown in Fig. 7b, the larger error of fine volume concentration with 42 % brought greater uncertainty to components estimation. Similarly, with the input CRI varying by more than an order of magnitude, Schuster et al. (2016) have found that the uncertainty of brown carbon can change from 50% to 440 %. Obviously, the estimation of BC was more sensitive to the input parameters. This may be

attributed to the smaller amount of BC, the volume fraction of which is one to two orders of magnitude less than that of other components. The uncertainties caused by the constraints of the extinction coefficients were mainly below 50 % for different components, which is comparable with the uncertainty of retrievals by remote sensing (Li et al., 2013). It should be noted that the uncertainty of aerosol components, such as BC in emission inventories, can be 200% and more (Schuster et al., 2005). Therefore, it's valuable to retrieve by our algorithm.

In fact, some errors exist exactly but are difficult to quantify in a realistic way. Just as the assumption of internal mixing doesn't apply to all situations, so do the microphysical parameters. Cheng et al. (2012) have observed that the number fraction of internally mixed soot in total soot particles had pronounced diurnal cycles. When the aging process converts externally mixed soot into internally mixed ones, emissions tend to emit more fresh and externally mixed soot particles. Another unquantifiable error is from the Mie theory based on the spherical hypothesis, which idealizes aerosol particles.

However, the uncertainties related to assumptions are endemic to all retrievals by remote sensing, as well as the chemistry transport models (Chen et al., 2019). We should mention that field measurement also cannot avoid inconsistent assumptions.



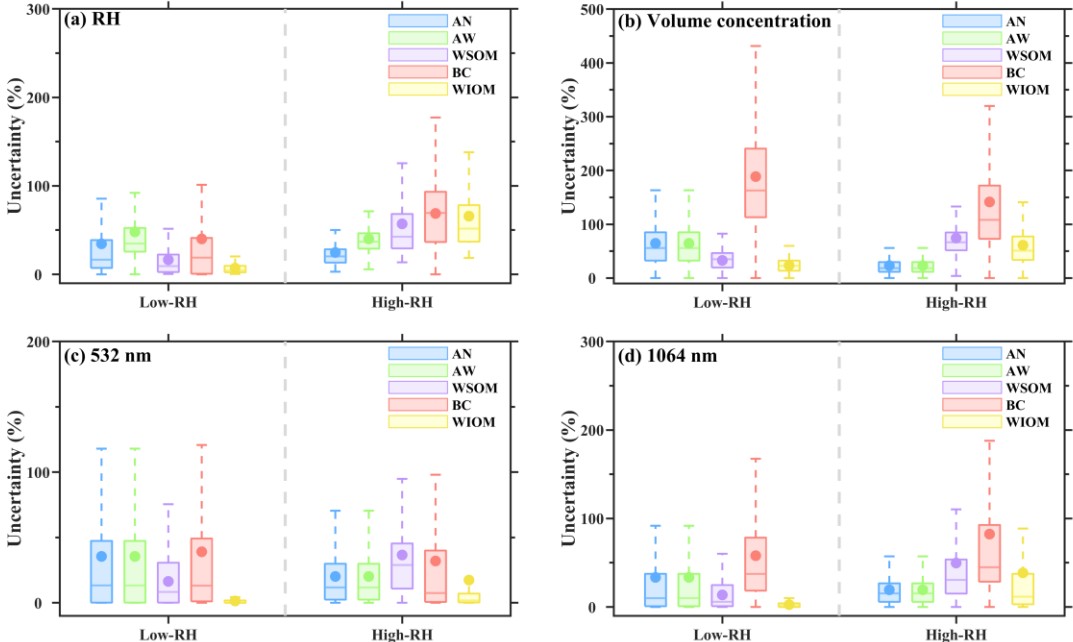

**Figure 7.** The uncertainty of component retrieval from (a) RH with the error of 10 %; (b) Volume concentration with the error of 42 %; extinction coefficient with the error of 14 % at (c) 532 nm; (d) 1064 nm.


## 3.3 The application of retrieval algorithm

### 3.3.1 Optical closure test

Based on the measurement data of lidar and sun-photometer in February of 2021, the mass concentration profiles of aerosol components in Beijing were retrieved. Figure 8 shows a quantitative optical closure test under different RH conditions to

validate the consistency between recovered extinction and the constraints from GARRLiC. It can be seen that the modeled extinctions at 532 nm and 1064 nm both had a good correlation with the reference values. The correlation coefficients were both close to 1. However, there were still some large residuals at the two wavelengths, especially at the RH between 70–80 %. It seems to underestimate the extinction when the RH was larger than 70 %. That's probably because the water-insoluble fraction is limited at high RH, and BC in water-insoluble matter tends to contribute greatly to extinction. On the

contrary, the overestimation of 532 nm at the RH of about 30 % can be attributed to the larger proportion of water-insoluble mater. We should realize that the parameterization scheme of water-soluble and water-insoluble matter may have trouble in reflecting the real atmosphere situation. But for now, there are still not enough observation experiments to construct a more realistic scheme. Moreover, the added constraint of the relationship between WIOM and WSOM can also limit the BC fraction. Although ignoring the constraint could bring about a well-matched closure result but might lead to unreasonable

component volume fractions.



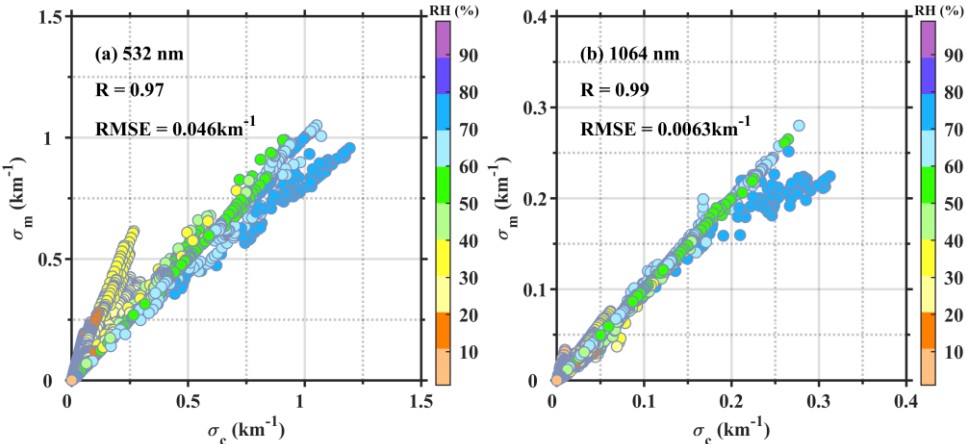

**Figure 8.** The comparison between modeled extinction $\sigma_m$ and extinction constraints from GARRLiC $\sigma_c$ at (a) 532 nm; (b) 1064 nm.

### 3.3.2 Vertical profiles of aerosol components in Beijing

Figure 9 shows two typical cases under different situations. It can be seen that the vertical distribution of aerosol components can be quantified even with extremely low aerosol loading (Fig. 9a). Under the clean condition, aerosols were mainly distributed below 3 km with different patterns of component profiles. There were similar peak values in the mass concentrations of AN and WIOM but at different heights. The mass concentration of AW was the smallest throughout the vertical direction due to the low RH. As shown in Fig. 9b, the two local maximums of RH being about 40 % appeared at about 700 m and 2.4 km respectively, which was consistent with extinction profiles below 3 km. Besides, there was a fairly good relationship between the optical fit extinctions and inputs from GARRLiC with a mean relative error of 4.63 % at 532 nm and -1.99 % at 1064 nm (Fig. 9c).

In the polluted case, aerosol components were concentrated below 1 km due to the weak atmospheric diffusion capacity as shown in Fig. 9d. The maximum mass concentration of AN occurred near the surface, being about 125 µg m$^{-3}$. Subsequently, there was a decreasing trend with a fluctuation between about 300 m to 600 m. While the mass concentration of WIOM and WSOM peaked at 672 m and were only about 10 µg m$^{-3}$ near the ground. According to Lou et al. (2017), sulfate and nitrate, transformed from $SO_2$ and $NO_2$, were mainly responsible for the fine particle pollution at the RH about 70 %, which explains the high proportion of AN in Fig. 9d. Generally, pollution is usually accompanied by high RH. As shown in Fig. 9e, the maximum RH matched the large value of extinction below 1 km well. Moreover, the well-recovered extinction profiles at two lidar wavelengths indicated the stability of our algorithm.


**Figure 9.** (a) The mass concentration profiles of aerosol components retrieved from remote sensing at 9:00 (LT) on 7 February 2021, which was under clean condition; (b) The vertical distribution of relative humidity (RH) (green line) and extinction coefficients at 532 nm at 9:00 (LT) on 7 February 2021. The red dotted line represents the extinction profile recovered by the components results and the dark grey line represents the input data from GARRLiC; (c) The vertical distribution of extinction coefficients at 1064 nm at 9:00 (LT) on 7 February 2021; (d)–(f) Same as (a–c) but for 12:29 (LT) on 26 February 2021, which was under polluted condition.





## 4. Conclusions

By combining ground-based lidar and sun-photometer, we develop a new algorithm to get the vertical profiles of fine mode aerosol components, including black carbon (BC), water-insoluble organic matter (WIOM), water-soluble organic matter (WSOM), ammonium nitrate-like (AN), and fine aerosol water content (AW), which increases the retrieved aerosol types
from dual-wavelength Mie lidar. Aided by the volume concentration profile from GARRLiC/GRASP, the optical mixing model used in the sun-photometer algorithm can be directly employed to construct extinction based on Mie theory. At the same time, hygroscopic growth is also considered to constrain AN and AW in the volume size distribution model. Then the residual between modeled extinction and constraints is quantified by a kernel function to find the optimal combination of component volume fractions. In this study, the vertical profiles of aerosol components in the February of 2021 in Beijing are
retrieved and compared with in situ measurements and simulated results from NAQPMS. There is the best consistency between the estimated and observed BC with a correlation coefficient up to 0.91. The trend of AN between estimation and observation is accordant but with a little underestimation. While the retrieved AN concentrations are more close to the simulated results (R=0.85). These results prove the validity of our components estimation. Besides, the reliability of the retrieval algorithm is also verified by the well-recovered extinction coefficients in the quantitative optical closure test.
Based on the products of AERONET, the evaluated mean errors of input parameters are introduced to assess the uncertainty of components estimation by Monte Carlo method. The uncertainties caused by extinction coefficients are mainly below 50 % for different components. And the more accurate input parameters are, the better estimated component results can be expected. However, the errors from the assumptions, such as internal mixing and spherical hypothesis, are difficult to quantify in a realistic way. We should mention that the assumptions are endemic to all retrievals by remote
sensing. On the other hand, the parameterization schemes and aerosol microphysical parameters used in the algorithm, which are variable over time and place, still need to be improved by enough observation experiments. In the future, the distinguishable aerosol types will increase by upgrading parameterization schemes, employing more lidar wavelengths, and considering the irregular shape of dust.

**Data availability.**

All data in this manuscript are freely available upon request through the corresponding author (tingyang@mail.iap.ac.cn).

**Supplement.**

Text S1: The verification of relative humidity from ERA5; Figure S1: The comparison of RH data between ERA5 and sound.

**Author contributions.**

Futing Wang and Ting Yang designed the whole structure of this work. Futing Wang analyzed the data and wrote the manuscript. Haibo Wang provided the components data from NAQPMS. Ting Yang, Xi Chen and Zifa Wang helped polish the manuscript.

**Competing interests.**

The contact author has declared that neither they nor their co-authors have any competing interests.

**Disclaimer.**

Publisher's note: Copernicus Publications remains neutral with regard to jurisdictional claims in published maps and institutional affiliations

**Acknowledgments.**

We would like to thank the support from the Strategic Priority Research Program of the Chinese Academy of Sciences (Grant No.XDA19040203). National High Technology Research and Development Program of China (No.2019YFC214802). The Young Talent Project of the Center for Excellence in Regional Atmospheric Environment, CAS (CERAE201803). The author Ting Yang gratefully acknowledges the Program of the Youth Innovation Promotion Association (CAS).

**Financial support.**

This work was supported by the Strategic Priority Re-search Program of the Chinese Academy of Sciences (Grant No. XDA19040203). National research program for key issues in air pollution control (DQGG202106).The author Ting Yang gratefully acknowledges the Program of the Youth Innovation Promotion Association (CAS).

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
