# Peer review of "Algorithm for vertical distribution of boundary layer aerosol components in remote sensing data"

_Atmospheric Measurement Techniques, 2022_

## Author Response (AR1)

**Authors' responses to Referees' comments**

**Journal:** Atmospheric Measurement Techniques

**Ms. Ref. No.:** amt-2022-165

**Title:** Algorithm for vertical distribution of boundary layer aerosol components in remote sensing data.

**Authors:** Futing Wang, Ting Yang, Zifa Wang, Haibo Wang, and Xi Chen, Yele Sun, JianJun Li, Guigang Tang, Wenxuan Chai
* * *
**Anonymous Referee #1**

This paper describes an algorithm to derive the profiles of fine mode aerosol components using lidar and sky radiometer data. GARRLiC is used to obtain fine mode extinction coefficient profiles at 532 nm and 1064 nm and fine mode volume concentration profile. Models for aerosol components including internal mixing are used to calculate complex refractive index and volume size distribution of the mixture of the aerosol components, and the extinction coefficients at 532 nm and 1064 nm are calculated by Mie scattering theory. The mixing ratios of the aerosol components are obtained by minimizing the residual between the calculated extinction coefficients and those from GARRLiC.

The method was applied to the data in Beijing and interesting results are reported. The analysis of errors are also provided. The method seems practically useful for analyzing profiles of aerosol components. Publication is recommended after revisions. The paper in the current form is difficult to read because the outline of the algorithm is not explained in the beginning of the methodology section. It is recommended to present Figure 1 in the beginning.

*Authors' response:*

We thank you very much for your suggestion. As you said, presenting Figure 1 in the beginning of the methodology section can help readers better understand the implementation of the

algorithm and changes have been made in the text. Figure 1 was moved to the end of section 2.1.1. Besides, the flowchart has also been polished to explain the algorithm more clearly.

Page 4, Line 110: "*Based on the distinct aerosol microphysical characteristics, we retrieve the fine-mode aerosol components profiles by constructing the aerosol model and microphysical parameterization schemes. Figure 1 gives the flowchart of our algorithm proposed in this study and the details will be described below.*"

[Figure]

*Figure 1. Flowchart of the algorithm proposed in this paper.*

**Authors' responses to Referees' comments**

**Journal:** Atmospheric Measurement Techniques

**Ms. Ref. No.:** amt-2022-165

**Title:** Algorithm for vertical distribution of boundary layer aerosol components in remote sensing data.

**Authors:** Futing Wang, Ting Yang, Zifa Wang, Haibo Wang, and Xi Chen, Yele Sun, JianJun Li, Guigang Tang, Wenxuan Chai
* * *
**Francisco Moler (Referee) #2:**

The manuscript titled "Algorithm for vertical distribution of boundary layer aerosol components in remote sensing data" by Wang et al., presents a new algorithm to derive vertical mass concentration profiles of fine aerosol components from ground-based lidar and sun-photometer data. The methodology is well-explained, providing details of the different steps of the process and the results are validated using experimental data (AERONET products and surface observations) and model results (NAQPMS components profiles). The manuscript is well-written and it presents a substantial contribution to scientific progress within the scope of Atmospheric Measurement Techniques. There are several issues and technical comments that can be improved, and some minor issues and typos to correct, highlighted in the attached file.

*Authors' response:*

We thank the Reviewer for the constructive comments, which we have addressed in detail. The itemized responses are listed below.

(1) Referee's comment:

The relationships among the different components (section 2.1.4) are not clearly explained, further details must be provided about the deduction of the AN, AW and BC components from the WIOM and WSOM (line 185 – 190), as well as the constraints applied to the WSOM and WIOM components. Also, these details must be included in Fig. 1, where the initial guess seems to apply to the five components. The manipulations required to compare these components with

surface measurements and model products, mentioned in lines 262 – 266 and lines 288 – 289, must be commented as well in this section, including some estimations of the errors produced by the conversions.

*Authors' response:*

We are sorry that we didn't explain clearly in the section 2.1.4. According to the Referee's comment, we improve the expression of this section and Fig. 1. The details about the deduction of different components are added and the subsequent manipulations are commented in section 2.1.4, which is marked in the manuscript.

Page 8, Line 218: "*Based on the above relationship, the fine mode aerosols are divided into two categories: water-insoluble aerosols and water-soluble aerosols, which can be quantified with the help of climatological parameterization scheme in Eq. (9). For water-insoluble aerosols (BC and WIOM), once the volume fraction of one is known, the other is determined. While for water-soluble aerosols (AN, AW, and WSOM), the relationship between AN and AW is constructed by Eq. (15), which can be considered as a whole. So only two unknowns, one from water-soluble and the other from water-insoluble species, are enough to get what we want. In our algorithm, we iteratively changed the volume fractions of WIOM and WSOM. Correspondingly, that of AN, AW, and BC can be obtained. What's more, we also constrained the relationship between WSOM and WIOM to ensure the reliability of inversion, . The ratio of WSOM mass concentration to the total OM is limited to 0.44 to 0.77, which has been applied in Zhang et al. (2018a) according to the statistics of observation experiments.*"

Page 9, Line 227: "*In summary, as presented in Fig. 1, if the volume fractions of WIOM and WSOM are initialized, the other species would be determined with the aid of parameterization schemes. Then the extinction coefficient can be calculated by the constructed aerosol model. Through multiple iterations and the constraints of fine mode extinction coefficients from GARRLiC, the optimal combination of volume fractions will be found. Subsequently, the optimal mass concentration results are compared with surface components measurements and model products, including OM, BC and AN, which verifies the inversion performance of our algorithm. Besides, the possible sources of error are discussed and the uncertainties from these sources are assessed in Sect. 3.2.*"

[Figure]

*Figure 1. Flowchart of the algorithm proposed in this paper.*

(2) Referee's comment:

The limitation to the fine mode must be better established. It seems that the authors don't consider contribution of dust to the fine mode and it isn't clear if the components cover all posible type of aerosols naturally found in the atmosphere?. Or at least those that can appear in the fine mode?. A better description of the components considered and how they compare with

those provided by surface measurements and NAQPMS model products must be included in section 2.2.3 and relevant parts of the manuscript.

*Authors' response:*

Thank you very much for the suggestion. We did ignore the description about the fine mode aerosol types. In fact, there are often different aerosol definitions and classification schemes focusing on some key components, which depends on the limited inputs and specific research purposes.We should note that the aerosol components in the remote sensing model are not completely equivalent to the chemical compositions traditionally, and there are some limitations in identifying compositions compared with surface chemical measurements. For example, distinguishing sulfate and nitrate seems to be beyond the scope of remote sensing due to their similar optical properties in light scattering, particle size and shape. Despite all this, the common remote sensing components, including black carbon (BC), brown carbon (BrC), dust, organic matter (OM), ammonium nitrate-like (AN), sea salt, and water uptake, are close to the species defined in Intergovernmental Panel on Climate Change (IPCC) 2013 (Boucher et al., 2013) and enough to satisfy the need for climate change and environmental monitoring. Generally, sea salt aerosol is considered as coarse particles and neglected in Beijing (Li et al., 2013). Dust and OM have similar light-absorbing characteristics due to the presence of hematite (Formenti et al., 2014). Therefore, they are usually separated by particle size since OM and dust are mostly present in fine and coarse mode aerosols, respectively (Schuster et al., 2016; Zhang et al., 2018a). Certainly, fine mode dust will also be taken into account with sufficient constraints. By assuming the dust volume concentration ratio of fine to coarse mode, Xie et al. (2017) further separated fine and coarse mode dust with the aid of spectra refractivity at four wavelengths, sphericity and single scatter albedos. In this paper, considering the limited lidar constraints (only two wavelengths), we only focus on fine mode aerosol and treat it as a mixture of five components like Zhang et al. (2018a): BC, OM (WIOM + WSOM), AN, and AW, and omit the presence of dust in fine mode to optimize the algorithm performance. The work about the dust identification can be carried out in coarse mode in the future. The above description has been supplemented in Sect. 2.1.1. And the description in Sect. 2.2.3 has also been polished in the manuscript.

Page 3, Line 79: "*Atmospheric aerosol is a complicated mixture of different components, which have different size distributions and complex refractive indexes. This makes it possible to separate aerosol components by remote sensing. But we should note that the aerosol components in the remote sensing model are not completely equivalent to the chemical compositions traditionally, and there are some limitations in identifying compositions compared with surface chemical measurements. For example, distinguishing sulfate and nitrate seems to be beyond the scope of remote sensing due to their similar optical properties in light scattering, particle size and shape. Despite all this, the common remote sensing components, including black carbon (BC), brown carbon (BrC), dust, organic matter (OM), ammonium nitrate-like (AN), sea salt, and water uptake, are close to the species defined in Intergovernmental Panel on Climate Change (IPCC) 2013 (Boucher et al., 2013) and enough to satisfy the need for climate change and environmental monitoring. In fact, there are often different aerosol definitions and classification schemes focusing on some key components, which depends on the limited inputs and specific research purposes. Generally, sea salt aerosol is considered as coarse particles and neglected in Beijing (Li et al., 2013). Dust and OM have similar light-absorbing characteristics due to the presence of hematite (Formenti et al., 2014). Therefore, they are usually separated by particle size since OM and dust are mostly present in fine and coarse mode aerosols, respectively (Schuster et al., 2016; Zhang et al., 2018a). Certainly, fine mode dust will also be taken into account with sufficient constraints. By assuming the dust volume concentration ratio of fine to coarse mode, Xie et al. (2017) further separated fine and coarse mode dust with the aid of spectra refractivity at four wavelengths, sphericity and single scatter albedos. Besides, they also defined OM as non-absorbing and Hydrophobic aerosols to separate from inorganic salt and absorbing carbon (BC and BrC). While Zhang et al. (2018a) divided the OM into two categories (WIOM and WSOM) to better model complex liquid systems, and BrC is considered as a part of WIOM.*"

Page 4, Line 98: "*In this paper, considering the limited lidar constraints (only two wavelengths), we only focus on fine mode aerosol and treat it as a mixture of five components like Zhang et al. (2018a): BC, OM (WIOM + WSOM), AN, and AW, and omit the presence of dust in fine mode to optimize the algorithm performance, although it seems to not cover all possible aerosol types in the atmosphere.*"

**References:**

Boucher, O., Randall, D., Artaxo, P., Bretherton, C., Feingold, G., Forster, P., Kerminen, V.-M., Kondo, Y., Liao, H., and Lohmann, U.: Clouds and aerosols, in: Climate change 2013: the physical science basis. Contribution of Working Group I to the Fifth Assessment Report of the Intergovernmental Panel on Climate Change, Cambridge University Press, 571-657, https://doi.org/10.1017/CBO9781107415324.016, 2013.

Formenti, P., Caquineau, S., Chevaillier, S., Klaver, A., Desboeufs, K., Rajot, J. L., Belin, S., and Briois, V.: Dominance of goethite over hematite in iron oxides of mineral dust from Western Africa: Quantitative partitioning by X-ray absorption spectroscopy, Journal of Geophysical Research: Atmospheres, 119, 12,740-712,754, https://doi.org/10.1002/2014JD021668, 2014.

Li, Z., Gu, X., Wang, L., Li, D., Xie, Y., Li, K., Dubovik, O., Schuster, G., Goloub, P., Zhang, Y., Li, L., Ma, Y., and Xu, H.: Aerosol physical and chemical properties retrieved from ground-based remote sensing measurements during heavy haze days in Beijing winter, Atmos. Chem. Phys., 13, 10171-10183, 10.5194/acp-13-10171-2013, 2013.

Schuster, G. L., Dubovik, O., and Arola, A.: Remote sensing of soot carbon - Part 1: Distinguishing different absorbing aerosol species, Atmos. Chem. Phys., 16, 1565-1585, 10.5194/acp-16-1565-2016, 2016.

Xie, Y. S., Li, Z. Q., Zhang, Y. X., Zhang, Y., Li, D. H., Li, K. T., Xu, H., Zhang, Y., Wang, Y. Q., Chen, X. F., Schauer, J. J., and Bergin, M.: Estimation of atmospheric aerosol composition from ground-based remote sensing measurements of Sun-sky radiometer, J. Geophys. Res.-Atmos., 122, 498-518, 10.1002/2016jd025839, 2017.

Zhang, Y., Li, Z., Sun, Y., Lv, Y., and Xie, Y.: Estimation of atmospheric columnar organic matter (OM) mass concentration from remote sensing measurements of aerosol spectral refractive, Atmos. Environ., 179, 107-117, 10.1016/j.atmosenv.2018.02.010, 2018.

Sect. 2.3.3: *"For comparable aerosol components, we used the sum of the water-soluble inorganic salt from surface measurements and NAQPMS products, such as sulfate, nitrate and ammonium, was used to compare with AN. Due to the limited available data, the mass concentration of OC multiplied by the conversion factor of 1.7 (Burki et al., 2020) was considered as the observed organics to compare with the total retrieved OM (WIOM+WSOM).*

*And estimated BC can be directly validated by the observation data and model products. In this study, in addition to the correlation coefficient (R), two statistics of root-mean-square error (RMSE) and normalized mean bias (NMB) were introduced to evaluate the algorithm performance, which can be expressed as follows:*

$$RMSE = \sqrt{\frac{\sum_{i=1}^{n}(X_r - X_o)^2}{n}} \qquad (17)$$

$$NMB = \frac{\sum_{i=1}^{n}(X_r - X_o)}{\sum_{i=1}^{n} X_o} \qquad (18)$$

*Where $X_r$ represents different aerosol components of BC, AN, and OM; and $n$ is the sample size; $X_o$ is the corresponding components from surface measurements and NAQPMS products. As an index to measure the deviation from true values (Wang et al., 2021), NMB > 0 indicates the overestimation of estimated results. The larger the value, the greater the overestimation.*"

(3) Referee's comment:

Some technical issues:

Figure 1 must be revised to follow standard flowchart rules and include all relevant information of the procedure

Figure 4 & 5 X-axis labels are hard to read, please provide better labeling. Also Fig4 c Y-axis is wrong, values should be 0-100 %, not 0-1.

*Authors' response:*

We are sorry for the mistake and thank the Reviewer's reminder very much. The Figure 1, Figure 4 & 5 has been modified as follows:

[Figure]

*Figure 1. Flowchart of the algorithm proposed in this paper.*

[Figure]

*Figure 4. (a) The fine volume concentration from GARRLiC and AERONET; (b) The fine mode extinction at 532 nm and 1064 nm; (c) The RH from ERA5 at the available time on 8–15 February 2021.*

[Figure]

*Figure 5. The comparisons between observation and estimation results from remote sensing of (a) AN; (b) BC; (c) OM at available time on 8–15 February 2021.*

(4) Referee's supplement comment:

Thank you for reviewing our paper so carefully. Since some of these supplement comments have been answered in detail above, I will only elaborate on some comments not mentioned as follows:

① It should be clearer what has been developed for the first time. The synergy of lidar and sunphotometer was already developed by GARRLiC, so it must be the application to the fine-mode. Please rephrase it more clearly.

*Authors' response:*

Indeed, "for the first time" in the Abstract refers that we firstly proposed the algorithm to get the vertical mass concentration profiles of fine-mode aerosol components based on GARRLiC. In order to avoid ambiguity, the related description has been modified in the manuscript.

Page 1, Line 13: "*Based on the synergy of ground-based lidar and sun-photometer in*

*Generalized Aerosol Retrieval from Radiometer and Lidar Combined data (GARRLiC), this paper developed a new algorithm to get the vertical mass concentration profiles of fine-mode aerosol components for the first time."*

② Add acronym meaning and description (OM = WSOM + WIOM)

*Authors' response:*

"OM" in the Abstract has been explained and the description OM = WIOM + WSOM has been added in the manuscript.

Page 1, Line 21: *"The estimated and observed BC concentration matched well with a correlation coefficient up to 0.91, while there was an evident overestimation of organic matter (OM = WIOM + WSOM, NMB=0.98)."*

③ What do the authors mean by "uniquely"?

*Authors' response:*

"uniquely" in "the establishment of uniquely distributed ground-based network" means that the ground-based network AERONET is unique and irreplaceable. To avoid ambiguity, we have removed this description.

Page 2, Line 48: *"Besides, the establishment of ground-based networks, e.g., AERONET, AD-Net, and MPLNET, also improves the spatial detection resolution."*

④ As mentioned in the abstract, it should be clearer what has been developed for the first time. The synergy of lidar and sunphotometer was already developed by GARRLiC, so it must be the application to the fine-mode. Please rephrase it more clearly. Do these components cover all posible type of aerosols naturally found in the atmosphere?. Or only those that can appear in the fine mode?

*Authors' response:*

As mentioned in the comment (4)①, "for the first time" refers that we firstly proposed the algorithm to get the vertical mass concentration profiles of fine-mode aerosol components based on GARRLiC. In order to avoid ambiguity, the related description has been modified in

the manuscript. Besides, the aerosol components mentioned in the last paragraph of Introduction only cover the possible type appearing in the fine mode. The reason has been explained in detail in the comment (2).

Page 3, Line 70: "*In this study, based on the synergy of ground-based Mie lidar and sun-photometer in Generalized Aerosol Retrieval from Radiometer and Lidar Combined data (GARRLiC), a new algorithm to get the vertical profiles of **fine-mode aerosol components**, including black carbon (BC), water-insoluble organic matter (WIOM), water-soluble organic matter (WSOM), ammonium nitrate-like (AN), and fine aerosol water content (AW), is proposed for the first time..*"

⑤  "complex refractive indexes (CRI hereafter)". Please include acronims the first time it appears in the text. Better include acronym meaning in text, not in caption.

*Authors' response:*

Thank you very much for your reminder. The caption of Table 1 has been modified and the meaning of "CRI" are explained in the text.

⑥  How do the authors take into account the fraction of dust that appears in the fine mode. Although it doesn't contribute much to the volume distribution, it does to the number distribution.

*Authors' response:*

As explained in the comment (2), we didn't consider the fine mode dust. Dust and OM have similar light-absorbing characteristics due to the presence of hematite (Formenti et al., 2014). Therefore, they are usually separated by particle size since OM and dust are mostly present in fine and coarse mode aerosols, respectively (Schuster et al., 2016; Zhang et al., 2018). Certainly, fine mode dust will also be taken into account with sufficient constraints. By assuming the dust volume concentration ratio of fine to coarse mode, Xie et al. (2017) further separated fine and coarse mode dust with the aid of spectra refractivity at four wavelengths, sphericity and single scatter albedos. In this paper, considering the limited lidar constraints (only two wavelengths), we have no way to separate fine mode dust for the time being and only focus on fine mode

aerosol and treat it as a mixture of five components like Zhang et al. (2018): BC, OM (WIOM + WSOM), AN, and AW, and omit the presence of dust in fine mode to optimize the algorithm performance.

**References:**

Formenti, P., Caquineau, S., Chevaillier, S., Klaver, A., Desboeufs, K., Rajot, J. L., Belin, S., and Briois, V.: Dominance of goethite over hematite in iron oxides of mineral dust from Western Africa: Quantitative partitioning by X-ray absorption spectroscopy, Journal of Geophysical Research: Atmospheres, 119, 12,740-712,754, https://doi.org/10.1002/2014JD021668, 2014.

Schuster, G. L., Dubovik, O., and Arola, A.: Remote sensing of soot carbon - Part 1: Distinguishing different absorbing aerosol species, Atmos. Chem. Phys., 16, 1565-1585, 10.5194/acp-16-1565-2016, 2016.

Xie, Y. S., Li, Z. Q., Zhang, Y. X., Zhang, Y., Li, D. H., Li, K. T., Xu, H., Zhang, Y., Wang, Y. Q., Chen, X. F., Schauer, J. J., and Bergin, M.: Estimation of atmospheric aerosol composition from ground-based remote sensing measurements of Sun-sky radiometer, J. Geophys. Res.-Atmos., 122, 498-518, 10.1002/2016jd025839, 2017.

Zhang, Y., Li, Z., Sun, Y., Lv, Y., and Xie, Y.: Estimation of atmospheric columnar organic matter (OM) mass concentration from remote sensing measurements of aerosol spectral refractive, Atmos. Environ., 179, 107-117, 10.1016/j.atmosenv.2018.02.010, 2018.

⑦ How do the authors explain the large uncertainties under high-HR conditions for non-hygroscopic aerosols?

*Authors' response:*

Generally, severe pollution is usually accompanied by high RH (Tie et al., 2017; Liu et al., 2018) and strong extinction. On the other hand, BC and WIOM, which belong to the non-hygroscopic aerosols, are considered as the absorbing and insoluble aerosols in our aerosol model. Under the polluted situation, BC and WIOM may contribute more to the total extinction than that under clean condition. However, the parameterization scheme used in our algorithm can bring larger error of $\frac{f_i}{f_s}$ under high RH (Zhang et al., 2018), which will cause a large change in the

percentage of insoluble-matter and soluble-matter. In the soluble-matter, the parameterization scheme about the ratio of AN to AW was also affected by RH, which may offset the effect of RH on $\frac{f_i}{f_s}$. While the insoluble-matter only depends on $\frac{f_i}{f_s}$ and may be more affected by RH.

**References:**

Liu, Q., Jia, X., Quan, J., Li, J., Li, X., Wu, Y., Chen, D., Wang, Z., and Liu, Y.: New positive feedback mechanism between boundary layer meteorology and secondary aerosol formation during severe haze events, Scientific Reports, 8, 6095, 10.1038/s41598-018-24366-3, 2018.

Tie, X., Huang, R.-J., Cao, J., Zhang, Q., Cheng, Y., Su, H., Chang, D., Pöschl, U., Hoffmann, T., Dusek, U., Li, G., Worsnop, D. R., and O'Dowd, C. D.: Severe Pollution in China Amplified by Atmospheric Moisture, Scientific Reports, 7, 15760, 10.1038/s41598-017-15909-1, 2017.

Zhang, Y., Li, Z., Sun, Y., Lv, Y., and Xie, Y.: Estimation of atmospheric columnar organic matter (OM) mass concentration from remote sensing measurements of aerosol spectral refractive, Atmos. Environ., 179, 107-117, 10.1016/j.atmosenv.2018.02.010, 2018.

⑧ What does this sentence mean?, maybe errors that are unavoidable?

*Authors' response:*

In addition to the errors caused by the uncertainty of input parameters, the aerosol model and parameterization schemes used in the algorithm are also important sources of error. Just as the assumption of internal mixing doesn't apply to all situations, so do the microphysical parameters. Cheng et al. (2012) have observed that the number fraction of internally mixed soot in total soot particles had pronounced diurnal cycles. When the aging process converts externally mixed soot into internally mixed ones, emissions tend to emit more fresh and externally mixed soot particles. Another unquantifiable error is from the Mie theory based on the spherical hypothesis, which idealizes aerosol particles. However, the uncertainties related to assumptions are endemic to all retrievals by remote sensing, as well as the chemistry transport models (Chen et al., 2019).

**References:**

Chen, C., Dubovik, O., Henze, D. K., Chin, M., Lapyonok, T., Schuster, G. L., Ducos, F., Fuertes, D., Litvinov, P., Li, L., Lopatin, A., Hu, Q., and Torres, B.: Constraining global aerosol emissions using POLDER/PARASOL satellite remote sensing observations, Atmos. Chem. Phys., 19, 14585-14606, 10.5194/acp-19-14585-2019, 2019.

Cheng, Y. F., Su, H., Rose, D., Gunthe, S. S., Berghof, M., Wehner, B., Achtert, P., Nowak, A., Takegawa, N., Kondo, Y., Shiraiwa, M., Gong, Y. G., Shao, M., Hu, M., Zhu, T., Zhang, Y. H., Carmichael, G. R., Wiedensohler, A., Andreae, M. O., and Poschl, U.: Size-resolved measurement of the mixing state of soot in the megacity Beijing, China: diurnal cycle, aging and parameterization, Atmos. Chem. Phys., 12, 4477-4491, 10.5194/acp-12-4477-2012, 2012.

⑨ This is more a description of the procedure than a conclusion. This section should include only the relevant conclusions of the work.

*Authors' response:*

Thank you for your suggestion. We have deleted the related description in the conclusion section.

⑩ It would be more clear if the results from the comparison with the in situ measurements and with simulated results are separated in the explanation.

*Authors' response:*

Thank you for your suggestion. We separated the comparison with the in situ measurements and with simulated results and rewrote the relevant description in conclusion section.

Page 21, Line 449: *"On this basis, the vertical profiles of aerosol components in the February of 2021 in Beijing are retrieved and compared with in situ measurements and simulated results from NAQPMS, which prove the validity of our components estimation. There is the best consistency between the estimated and observed BC with a correlation coefficient up to 0.91. The trend of AN between estimation and observation is accordant but with a little underestimation. While compared with the simulated results, the retrieved AN from remote sensing had the best correlation (R=0.85) and there was a slight underestimation (NMB=-0.19).*

*The correlations of BC and OM were weaker relatively with a correlation coefficient of about 0.5. Vertically, the distribution of different aerosol components was synchronous in both remote sensing and NAQPMS. Considering the distinct principles, the differences between remote sensed and simulated results are resonable to some extent."*

Yele Sun provided constructive suggestions for the revision of the paper; JianJun Li, Guigang Tang and Wenxuan Chai provided aerosol components observation data to verify and improve the algorithm.

Yele Sun    sunyele@mail.iap.ac.cn

JiJuan Li    lijj@cnemc.cn

Guigang Tang    tanggg@cnemc.cn

Wenxuan Chai    chaiwx@cnemc.cn

---

## Author Response (AR2)

**Authors' responses to Referees' comments**

**Journal:** Atmospheric Measurement Techniques

**Ms. Ref. No.:** amt-2022-165

**Title:** Algorithm for vertical distribution of boundary layer aerosol components in remote sensing data.

**Authors:** Futing Wang, Ting Yang, Zifa Wang, Haibo Wang, Xi Chen, Yele Sun, JianJun Li, Guigang Tang, and Wenxuan Chai

The manuscript titled "Algorithm for vertical distribution of boundary layer aerosol components in remote sensing data" still has some minor issues, please revise the manuscript according to Referees' comments.

*Authors' response:*

We have revised the manuscript according to Referees' comments. The revisions have been marked in the track-changes manuscript file.